# A Reproduction of Ensemble Distribution Distillation

## Reproducibility Summary

**Scope of Reproducibility**

The authors claim that their proposed method is able to, given an ensemble of deep neural networks, capture the uncertainty estimation and decomposition capabilities of the ensemble into a single model. The authors also claim that this only results in a small reduction in classification performance compared to the ensemble. Most of the authors' experiments on the CIFAR-10 dataset were reproduced.

**Methodology**

The proposed method was re-implemented in `tf.keras`. The surrounding data pipelines, pre-processing, and experimentation code were also re-implemented. As in the original paper, the models were based on VGG-16 networks with random initialization, but trained within the project. Training and evaluation was done on two consumer-grade GPUs, for a total of 273 hours.

**Results**

Our findings support the authors' central claims. In terms of uncertainty estimation our $EnD^2$ achieved $(99 \pm 1)$ % of the AUC-ROC of our ensemble on the OOD-detection task. The corresponding value in the original paper was $(100 \pm 1)$ %. In terms of classification our $EnD^2$ had $(16 \pm 1)\%$ higher error than our ensemble. The corresponding values in the original paper was $(11 \pm 6)\%$. Other metrics showed similar agreement, but, significantly, in the OOD-detection task our EnD performed at least as well as our $EnD^2$. This is in stark contrast with the original paper.

We also took a novel approach to visualizing the uncertainty decomposition by plotting the resulting distributions on a simplex, offering a visual explanation to some surprising results in the original paper, while mostly supporting the authors' intuitive justifications for the model.

**What was easy**

The original paper features a thorough mathematical formulation of the method, aiding conceptual understanding. The datasets used by the authors are publicly available. The use of the simpler datasets also meant that it was computationally feasible for us to reproduce these results. The base model used is well known with several implementation available, allowing us to focus on the novel aspects of the method.

**What was difficult**

While the theoretical explanations of the method are excellent, we initially found it hard to translate this into an implementation. Our difficulty was likely caused by our inexperience with the subject matter. Nonetheless, a pseudocode, such as the one we have provided, would havee simplified the re-implementation. We were not able to reproduce the results on some of the datasets due to limited computational resources.

**Communication with original authors**

We did not contact the original authors directly, but we did refer to a public GitHub and blog post created by one of the authors. At the same time as submitting this report to the ML Reproducibility Challenge 2020 we also sent a copy to the authors and asked for their feedback.

# 1 Introduction

Uncertainty estimation can help to make deep learning safer and more usable by allowing the model to identify cases it is not suitable to handle. There are different kinds of uncertainty, however, and it is especially interesting to separate uncertainty caused by ambiguities or contradictions in the data from the uncertainty that arises when a model faces a situation it has not been trained for. Ensemble-based methods of uncertainty estimation are capable of making this distinction but suffer from computational requirements at the evaluation phase [1]. The authors of *Ensemble Distribution Distillation* (EnD$^2$) [2] address this issue by using the output of an ensemble to train a so-called Prior Network (PN) [3], distilling the ensemble down to a single model while also preserving its uncertainty decomposition abilities. This can be contrasted with regular ensemble distillation models [4] (EnD), which are not able to decompose uncertainty. The reproduced paper was accepted to ICLR2020.

# 2 Scope of reproducibility

We consider the setting of using CIFAR10 [5] as an in-distribution dataset, and LSUN [6] as an out-of-distribution dataset. Our supplementary material also examines the setting of using a synthetic dataset in $\mathbb{R}^2$ for visualization.

The claims from the original article that this reproduction is testing are as follows:

1. **Classification performance**: In terms of error rate, prediction rejection rate, and negative log-likelihood EnD$^2$ has worse performance than the ensemble, but similar performance to EnD and PriorNet, and better performance than the individual model. In terms of expected calibration error, EnD$^2$ has worse performance than the ensemble, but better performance than the other methods. On CIFAR-10 in particular, EnD$^2$ has the best expected calibration error of all models. This claim corresponds to Table 3 in the original paper.

2. **Out-of-distribution detection performance**: In terms of AUC-ROC on CIFAR-10 vs. LSUN, EnD$^2$ without auxiliary dataset performs worse than the ensemble and the PriorNet, similar to the individual model, and better than EnD. With the auxiliary dataset, however, EnD$^2$ performs as well as the ensemble, almost as well as PriorNet, and better than EnD. Using knowledge uncertainty as opposed to total uncertainty on CIFAR-10 vs. LSUN does not yield an improved AUC-ROC. This claim corresponds to Table 4 in the original paper.

3. **Dependency on ensemble size**: Using 20 models in the ensemble does better than using 5 models, but there is no conclusive gain when using more than 20 models.

4. **Dependency on temperature**: It is necessary to use temperature of at least 5 to successfully distribution-distill the ensemble. Using higher initial temperatures do not result in conclusive improvement.

5. **Uncertainty decomposition**: EnD$^2$ trained with an auxiliary dataset is able to reconstruct the uncertainty decomposition made possible by ensembles.

We reproduce all experiments of the main article and most of the appendix, except for the use of CIFAR100 and Tiny Imagenet datasets. Some of these results can be found in our supplementary material. From their appendix, we do not reproduce Table 7 in appendix B. We did not recreate the OOD-detection plots when reproducing the ablation study.

# 3 Methodology

## 3.1 Model description

We consider the same seven models as the original authors:

- IND: A single classification model.
- ENSM: An ensemble of independently trained IND models.
- EnD: A single model distilling ENSM trained according to [4].
- EnD$^2$: A single model distribution-distilling ENSM trained according to [2].
- EnD$_{+\text{AUX}}$: Like EnD, but trained with auxiliary data.
- EnD$^2_{+\text{AUX}}$: Like EnD$^2$, but trained with auxiliary data.
- PN$_{+\text{AUX}}$: A PriorNet model with auxiliary data trained according to [3]

These models are all based on almost identical VGG16 architectures [7], adapted to CIFAR-10 data as in [3] by adding dropout, batch normalization and reducing the size of the fully connected layers. The only exception being that batch normalization is not used for PN.

Table 1: Datasets used in the CIFAR-10 setting

| Dataset | No. samples | No. classes | Image dimensions | Link |
|---|---|---|---|---|
| CIFAR-10 train | 50000 | 10 | 32x32x3 | https://www.cs.toronto.edu/ kriz/cifar.html |
| CIFAR-100 train | 50000 | 100 | 32x32x3 | https://www.cs.toronto.edu/ kriz/cifar.html |
| CIFAR-10 test | 10000 | 10 | 32x32x3 | https://www.cs.toronto.edu/ kriz/cifar.html |
| LSUN test | 10000 | 10 | 256x256x3 | https://www.yf.io/p/lsun |

Table 2: Training parameters in the CIFAR-10 setting

| Model | Epochs | Cycle len. | $\eta_0$ | $\eta_{max}$ | $\eta_{min}$ | Dropout | $T_0$ | Annealing | AUX data |
|---|---|---|---|---|---|---|---|---|---|
| DNN | 45 | 30 | $10^{-3}$ | $10^{-2}$ | $10^{-6}$ | 0.5 | - | - | - |
| EnD | 90 | 60 | $10^{-3}$ | $10^{-2}$ | $10^{-6}$ | 0.7 | 2.5 | No | - |
| EnD$_{+AUX}$ | 90 | 60 | $10^{-3}$ | $10^{-2}$ | $10^{-6}$ | 0.7 | 2.5 | No | CIFAR-100 |
| EnD$^2$ | 90 | 60 | $10^{-3}$ | $10^{-2}$ | $10^{-6}$ | 0.7 | 10 | Yes | - |
| EnD$^2_{+AUX}$ | 90 | 60 | $10^{-3}$ | $10^{-2}$ | $10^{-6}$ | 0.7 | 10 | Yes | CIFAR-100 |
| PN | 45 | 30 | $0.5 \cdot 10^{-3}$ | $0.5 \cdot 10^{-2}$ | $0.5 \cdot 10^{-6}$ | 0.7 | - | No | CIFAR-100 |

## 3.2 Dataset

The training set of CIFAR-10 was used as the primary training dataset. The training set of CIFAR-100 was used as an auxiliary dataset. For evaluating the classification task the test set of CIFAR-10 was used. For evaluating the out-of-distribution detection task the CIFAR-10 test set was used as in-domain dataset, while the LSUN test set was used as the out-of-domain dataset. Information about the datasets is listed in Table 1.

Each image $x$ was normalized according to $x' = x/127.5 - 1$ where the operations are elementwise, causing all values to lie in the range (-1, 1). The LSUN images were also scaled down to 32x32. Furthermore, dataset augmentation was used for all models, consisting of rotations with $15°$ range, horizontal flips, width and height shifts of up to 4 pixels in each direction, and using nearest-neighbour interpolation.

## 3.3 Hyperparameters

The models were trained with the hyperparameters listed in Table 2.

## 3.4 Experimental setup and code

Using these models and dataset we ran a number of experiments, as detailed below. The full code is available on https://anonymous.4open.science/r/4ee2c9ef-295f-44e2-8214-f0818b932817/. Our implementation was made in TensorFlow Keras, as opposed to the original implementation which was made in PyTorch.

**Classification:** The classification task was evaluated on the test set of CIFAR-10. We use the same four metrics as in the original paper, ERR, PRR, ECE, and NLL. ERR is the mean classification error. PRR is the prediction rejection area ratio introduced in Appendix B of [2]. ECE is the expected calibration error[1]. Finally, NLL is the negative log-likelihood. This experiment tests Claim 1.

**Out-of-distribution detection:** The OOD-detection task was evaluated with the CIFAR-10 test set as the in-domain set, and the LSUN test set as the out-of-domain set. The AUC-ROC was computed both when total uncertainty and when only knowledge uncertainty is used to make rejection decisions. This experiment tests Claim 2.

**Ensemble size ablation study:** Our examination of the effect of ensemble size goes slightly beyond the original authors. We extend the error analysis to also consider the sensitivity of EnD$^2$ to variations in the underlying ensemble. We began by training a set of 400 VGG16 models on CIFAR-10. Next, we sampled randomly from this set to create 4 different sets, each consisting of 100 models. For each $N \in \{1, 2, 3, 4, 6, 8, 10, 13, 16, 20, 25, 30, 45, 60, 75, 100\}$ we trained four EnD$^2$ models on an ensemble consisting of the first $N$ models in the first of the four sets, corresponding to what was done in the original study. We also trained *one* model on an ensemble consisting of the first $N$ models *for each* of the three remaining sets, capturing the sensitivity of EnD$^2$ to changes in the underlying ensemble. All ensemble and EnD$^2$ models were then evaluated on the classification task. This experiment tests Claim 3.

---

[1] We used the open-source implementation in https://github.com/google/uncertainty-metrics.

Table 3: Classification metrics on CIFAR-10. Error bounds signify two standard deviations, taken over three models. Up-arrow (↑) indicates that higher is better, down-arrow (↓) indicates that lower is better.

| Crit. | IND | ENSM | EnD | $EnD^2$ | $EnD_{+AUX}$ | $EnD^2_{+AUX}$ | $PN_{+AUX}$ |
|---|---|---|---|---|---|---|---|
| ERR↓ | $9.87_{\pm0.70}$ | $8.80_{\pm NA}$ | $\mathbf{8.70}_{\pm0.53}$ | $9.90_{\pm0.20}$ | $9.90_{\pm0.20}$ | $10.17_{\pm0.12}$ | $10.00_{\pm0.35}$ |
| PRR↑ | $69.80_{\pm1.31}$ | $\mathbf{80.30}_{\pm NA}$ | $78.67_{\pm0.12}$ | $76.97_{\pm0.83}$ | $78.37_{\pm1.21}$ | $77.20_{\pm0.72}$ | $56.57_{\pm9.49}$ |
| ECE↓ | $68.18_{\pm0.57}$ | $1.65_{\pm NA}$ | $\mathbf{1.56}_{\pm0.09}$ | $2.39_{\pm0.22}$ | $1.77_{\pm0.31}$ | $3.04_{\pm0.49}$ | $9.37_{\pm0.62}$ |
| NLL↓ | $1.58_{\pm0.01}$ | $\mathbf{0.25}_{\pm NA}$ | $0.26_{\pm0.01}$ | $0.33_{\pm0.00}$ | $0.29_{\pm0.00}$ | $0.34_{\pm0.00}$ | $0.46_{\pm0.00}$ |

Table 4: OOD AUC-ROC↑ on CIFAR-10 (in) and LSUN (out). Error bounds signify two standard deviations, taken over three models. Up-arrow (↑) indicates that higher is better, down-arrow (↓) indicates that lower is better.

| Unc. | IND | ENSM | EnD | $EnD^2$ | $EnD_{+AUX}$ | $EnD^2_{+AUX}$ | $PN_{+AUX}$ |
|---|---|---|---|---|---|---|---|
| Tot. | $86.63_{\pm0.31}$ | $90.00_{\pm NA}$ | $89.87_{\pm0.46}$ | $88.33_{\pm0.42}$ | $90.60_{\pm0.20}$ | $90.23_{\pm0.12}$ | $\mathbf{92.03}_{\pm0.46}$ |
| Know. | - | $89.30_{\pm NA}$ | - | $84.70_{\pm1.25}$ | - | $88.07_{\pm0.46}$ | $\mathbf{90.97}_{\pm0.42}$ |

**Temperature ablation study**: We reproduce the temperature ablation study by training $EnD^2$ models for various initial temperatures. For each $T \in \{1, 2, 3, 4, 5, 7.5, 10, 15, 20\}$ we trained three $EnD^2$ models with initial temperature $T$ on an ensemble consisting of 100 VGG16 models. The $EnD^2$ models were then evaluated on the classification task. In this experiment, we have chosen to use a slightly finer spacing between the temperatures than what the original authors used. This experiment tests Claim 4.

**Simplex visualization:** A key motivation for $EnD^2$ is the idea that an ensemble can distinguish between knowledge uncertainty and data uncertainty, and that this distinction is retained by the $EnD^2$ model. This is communicated using a schematic figure showing ensemble predictions on a simplex. A similar schematic figure can be found in [3], depicting a Dirichlet PDF of a PriorNet on a simplex. We recreated these figures using experimental data in order to examine Claim 5 from a novel perspective. A new training set was created, consisting of all images from the CIFAR10 train set with one of three labels chosen for their similarity: 'deer', 'horse', and 'dog'. The remaining images were reserved as out-of-distribution dataset for testing. CIFAR-100 was used as auxiliary data. An ensemble and $EnD^2$ was then trained on this data using the same architecture and processed as before. We then selected various images from the test set and visualized the ensemble predictions as well as the PDF of the $EnD^2$ model. The simplex visualization was created using open source code [2].

## 3.5 Computational requirements

Training and evaluation were performed on two mid-range consumer GPUs (RTX 2070, GTX 1660s) locally. Regarding VRAM, at least 4711 MiB is required for the models. The total number of GPU time required for the final results is 11.4 GPU days on an RTX 2070. The accumulated GPU days during the reproduction is 3-5 times this amount. We provide detailed numbers in the supplementary materials.

# 4 Results

## 4.1 Classification performance

The classification results are shown in Table 3. Overall, the ensemble seems to perform best, and when it does not, it is still within error bounds. Curiously, $EnD^2_{+AUX}$ seems to perform worse than the individual model in regards to ERR.

## 4.2 Out-of-distribution detection performance

The OOD-detection results are shown in Table 4. The results suggest plain $EnD^2$ performs worse than ENSM, but that the addition of an auxiliary dataset brings the performance up to at least the level of ENSM. More surprising, perhaps, is that $EnD^2$ seems to perform worse than EnD. In both metrics $PN_{+AUX}$ has a significant lead. Using knowledge uncertainty instead of total uncertainty decreases the effectiveness of all tested models. The supplemental material contains histograms showing the distribution of estimated total and knowledge uncertainty over the images.

---

[2]http://blog.bogatron.net/blog/2014/02/02/visualizing-dirichlet-distributions/

### 4.3 Ensemble size ablation study

Figure 1 shows the results of the ensemble size ablation study. The lines 'ENSM Paper' and 'EnD$^2$ Paper' show the results of the original paper. The bands indicate two standard deviations. Two bands surround the 'EnD$^2_{+AUX}$' line, representing the two types of variation we have examined. The purple band represents the variation of four EnD$^2$ models each trained on a different ensemble. The orange band represents the variation of four EnD$^2$ models all trained on the same ensemble. The band surrounding the 'EnD$^2_{+AUX}$ Paper' line corresponds to the latter type of variation.

There appears to be a trend of small improvement when the number of models is increased, but the high level of uncertainty makes it difficult to draw conclusions from the remaining points. Nonetheless, the results seem to generally indicate that EnD$^2$ is not particularly sensitive to ensemble size.

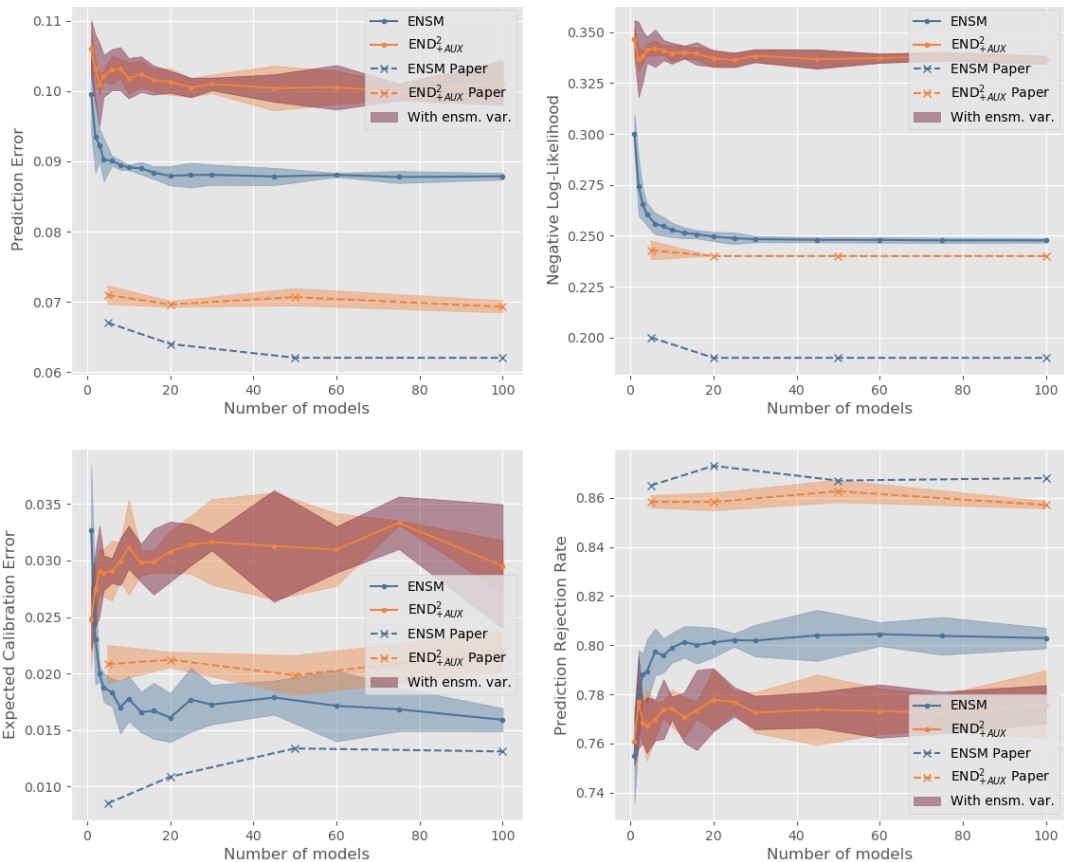

Figure 1: Ensemble size ablation study on CIFAR-10 classification.

### 4.4 Temperature ablation study

The results of our temperature ablation study are shown in Figure 2, along with the results of the original paper. For initial temperature equal to 1 and 2 our models fail to converge, resulting in poor classification performance. Raising the initial temperature to 3 allows the model to converge. Increasing the initial temperature further has no significant effect.

It is worth noting the negative PRR values for $T = 2$. The original authors mention this possibility when they propose the metric, and offer the interpretation that this means that the model is *increasing* the classification error by rejecting samples, performing worse than simply rejecting at random.

### 4.5 Simplex visualization

Predictions for four images are visualized in Figure 3. These four images were selected from the CIFAR10 dataset for respectively having the lowest total uncertainty, highest data uncertainty, highest knowledge uncertainty, and highest total uncertainty, as measured by the ensemble. The third row shows the Dirichlet PDF of EnD$^2$. There is a strong tendency towards extremely sharp distributions, even when the ensemble has high spread, making comparison difficult.

For this reason the fourth row plots the PDF after being transformed by the transformation $log(x+1)$. It is now possible to see that the PDF is adapting to the distribution of the ensembles.

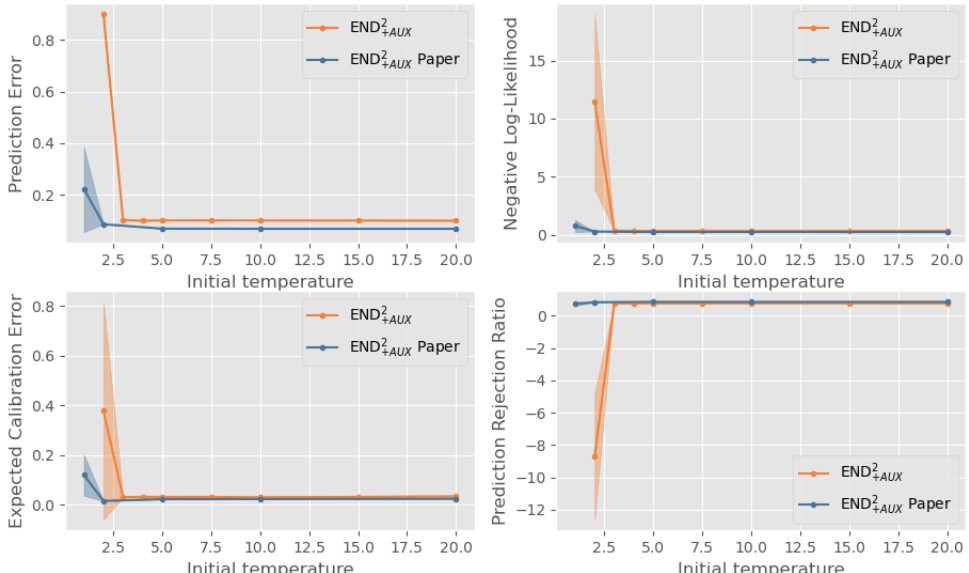

Figure 2: Temperature ablation study on CIFAR-10 classification.

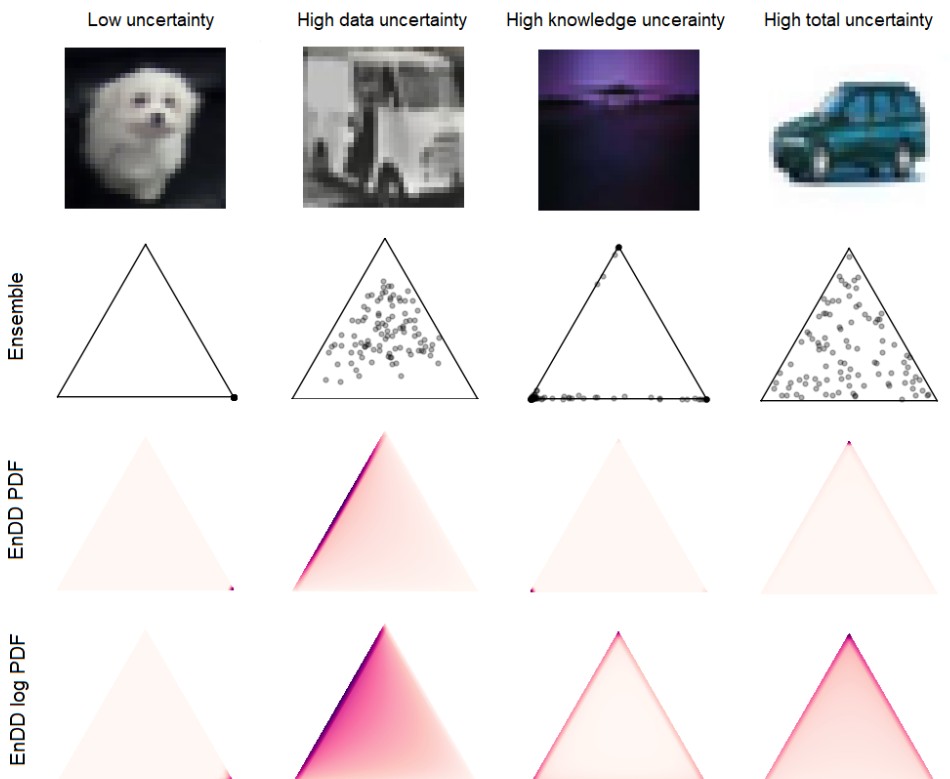

Figure 3: Visualization of ensemble distribution and EnD$^2$ PDF. The classes are, from left, to top, to right, Deer, Horse and Dog.

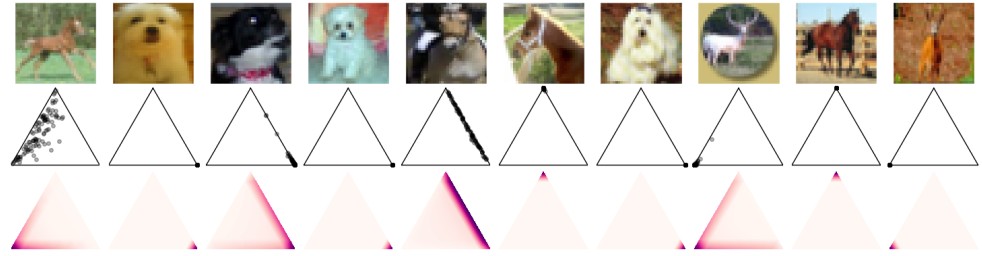

Figure 4: Random images from in-domain.

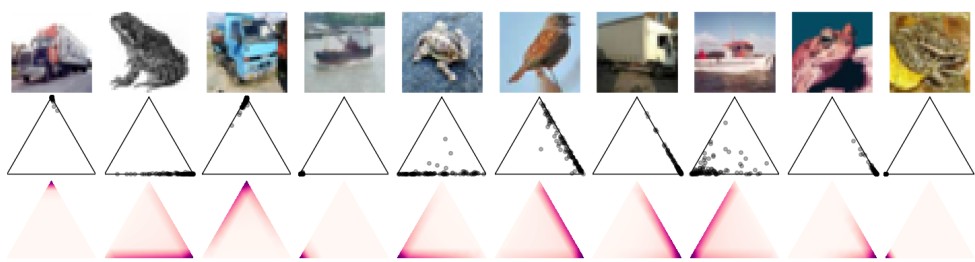

Figure 5: Random images from out-of-domain.

We also plot randomly selected images from the in, out, and auxiliary datasets respectively. The PDF has again been transformed using $log(x + 1)$. Figure 4 shows images from the in-domain dataset, and Figure 5 shows images from the out-of-domain dataset. The PDF appears to follow the ensemble fairly well, but it is noteworthy that the ensembles show such a low degree of spread despite encountering samples on which they have not been trained.

## 5 Discussion

### 5.1 Comparison with original paper

We now revisit the six claims which we specified in Section 2.

1. **Classification performance:** When compared to the original table we see overall worse performance. This is likely rooted in the fact that we were unable to achieve as high accuracy on our base VGG16 as in the original article. We therefore instead consider the relative performance between the models. Our supplementary material contains a table allowing for easy comparison with the original results. For example, we find that our EnD$^2$ has 112.5% of the classification error of the ensemble, while in the original paper this figure is 117.7%. The absolute difference is the same in both papers, 1.1 percentage units. Our results generally agree well, with those of the authors. There are some discrepancies in expected calibration error, but our extremely high ECE for the individual model suggests that there might be an issue in our computations of this metric. Overall our findings support Claim 1.

2. **Out-of-distribution detection performance**: For the most part, our results agree with Claim 2. For instance, we found that using total uncertainty EnD$^2$ without auxiliary data had 98.1% of the AUC-ROC of the ensemble, while the corresponding figure with auxiliary data was 100.0%. In the original paper, these figures were 96.8% and 99.8% respectively. There is one very significant discrepancy, however. With auxiliary dataset, our EnD$^2$ had 99.6% of the AUC-ROC of our EnD, while in the original paper this figure is 106.5%. A similar relationship exists without the auxiliary dataset. It is worth noting that in the original paper EnD performs worse than even the individual model, and the authors themselves note that this is odd. Since EnD$^2$ is designed to overcome certain shortcomings of EnD in terms of uncertainty estimation we believe that this warrants further investigation.

3. **Dependency on ensemble size**: For prediction error and negative log-likelihood, our results confirm the relative performance between ensembles and EnD$^2_{+AUX}$, with increased resolution. For expected calibration

error, the relative performance is confirmed for a large number of models, but for a small number of models, we get contradictory results. Their results seem to suggest that smaller ensembles have worse calibration, which is not expected, as per [1]. Our results confirm this expectation. In their paper, they state this expectation, but we see no comment for this discrepancy. For prediction rejection rate, we confirm the relative performance, and also show that it starts to drop rapidly below their tested range.

4. **Dependency on temperature annealing**: Our results diverge heavily from the results in the paper for temperatures 1 and 2. While the original authors are able to train working but sub-par models with these temperatures, we are unable to get the models to converge at all. We re-did the experiments with a new ensemble, and experimented with the smoothing factor and auxiliary data, but were unable to find any explanation for this difference. Nevertheless, these findings support the claim that temperature annealing is essential for successful use of the $EnD^2$ method. The authors suggested temperature 5 as a minimum value beyond which larger values make no difference. Our findings support this as well, although our increased resolution reveals that the minimum value for the CIFAR-10 dataset is closer to 3 than 5.

5. **Uncertainty decomposition**: Based on the description in [3] an image with a high knowledge uncertainty should produce a Dirichlet PDF with a close to uniform spread. Our simplex visualizations on the 3-class+AUX dataset shows that this is not the case. This is not too surprising, given that high knowledge uncertainty correlates with small alphas, and this in turn produces convex as opposed to flat probability density surfaces. Overall, these plots suggest that $EnD^2$ can capture the uncertainty decomposition of the ensemble.

The plots also show an interesting behaviour in the ensemble. The ensembles agree to a surprising extent on the out-of-domain samples. In fact, when they do disagree it normally takes the form of data uncertainty as opposed to knowledge uncertainty. This could perhaps shed some light on the observation that knowledge uncertainty does not seem to be useful for OOD-detection on CIFAR-10. The original authors explain this as essentially being a property of the dataset. We feel, based on the visualizations, that another possibility might be that the ensemble models simply are not diverse enough to provide a useful measure of knowledge uncertainty.

## 5.2 What was difficult

Although the general idea of the paper is well formulated in mathematical terms, the original paper does not provide many hints regarding how to implement the method. In our case, this imposed a significant barrier to immediately reproducing the work, since our inexperience meant that we're unable to immediately see how it could be implemented in a modern deep learning framework. There is some code available in a public repository hosted by one of the authors but this is not mentioned in the paper, and so we could not treat it as an official implementation. We have provided a pseudocode in our supplemental material, in order to hopefully assist future reproducers.

There are also some missing details regarding the models used. Most importantly, the authors mention that they have used a modified VGG model, but do not specify what these modifications are. The authors also do not specify the min and max value of the cyclic LR. These details may explain the consistently worse performance of our models despite the attempt of replication.

## 5.3 What was easy

The synthetic dataset was fairly easy to reconstruct, and the other datasets are well known and publicly available. The data augmentation was straightforward and easy to incorporate into a training pipeline. The base model (VGG16) used in most of the experiments is well known and was computationally feasible to train. Similarly, the datasets are not excessively demanding in terms of computation, although in our case training time did become a limiting factor due to the amount of time we spent on implementation and experimentation. The mathematical formulation of the model is very good, helping the conceptual understanding.

## 5.4 Communication with original authors

We did not communicate with the authors while reproducing their work, although we did refer to some resources which one of the authors has made publicly available, including an repository [3] made for [3] containing an implementation of $EnD^2$. At the same time as submitting this report, we also sent a copy to the authors and asked for their comments.

---

[3] https://github.com/KaosEngineer/PriorNetworks/tree/master/prior_networks

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
