# OpenReview forum: "A Reproduction of Ensemble Distribution Distillation"
_ML_Reproducibility_Challenge/2020 — RC2020_

### Official Review · AnonReviewer1 · 2021-03-01
**Review of "A Reproduction of Ensemble Distribution Distillation"**

**Rating:** 7
**Confidence:** 5

**Review:**

Summary:
This work reproduces the results  for "Ensemble Distribution Distillation" which uses a Dirichlet parametrization to distill deep ensembles to preserve uncertainty. The reproducibility report shows an extensive evaluation of CIFAR-10, ablation studies on ensemble size and temperature annealing and also includes visualization for uncertainty.

Strengths:
* Detailed reproduction Ensemble Distribution Distillation for CIFAR-10
* Report included also all baseline models which offers also relative comparisons
* Nice simplex visualization for classification experiments.

Weaknesses:
* I only have comments about clarity of the paper below.

Detailed comments and questions:
* Reproducibility summary: The scope of reproducibility is rather a description of the claims of the paper and is a bit vague. From my understanding, this should describe the scope of this work.
* Reproducibility summary: "Most of the authors' experiments on the CIFAR-10 dataset" were reproduced -> I think this should rather go to 'scope of reproducibility'. Here, it would be also good to mention any pre-trained models that you used, e.g. VGG16.
* Section 2 (Scope of reproducibility): What is not included in this work? Which experiments were not run that were included in the original paper?
* Claims 3-5: What does the original paper claim here in comparison?
* Table 3 & 4: Arrows up and down indicating "higher is better" or "lower is better" would be useful for all the metrics shown.

**Familiar With The Original Paper:**

I have read the original paper

**Reproducibility Summary:**

Report has summary

---

### Official Review · AnonReviewer3 · 2021-03-02
**The reviewer did a great job reviewing and reporting details of this paper.**

**Rating:** 8
**Confidence:** 3

**Review:**

Introduction is quite intuitive, giving a high-level context to the paper being reviewed.
The Scope of reproducibility was well highlighted,clear and concise.
All claims identified were supported by experiments.
Although reviewer reproduced the paper using Tensorflow keras contrary to the original paper, The results obtained was still similar to the original paper, although not 100%.

The overall reproduced paper is concise, explanatory and of good quality.

**Familiar With The Original Paper:**

I have read the original paper

**Reproducibility Summary:**

Report has summary

---

### Decision · Program_Chairs · 2021-03-31

**Decision:**

Accept

**Comment:**

Super complete, well-implemented and well-explained reproducibility report, with an in-depth reflection regarding the original paper.